# Analysis of Circulation Characteristics and Heat Balance of High-Speed Rolling Bearing under Oil-Air Lubrication

Xiqiang Ma [1,2,3], Mian Zhang [1], Fang Yang [1,2,3], Yujun Xue [1,2,3], Ruijie Gu [1,2,3] and Nan Guo [1,3,*]

1 School of Mechatronics Engineering, Henan University of Science and Technology, Luoyang 471003, China
2 Innovative Research Team of Livestock Intelligent Breeding and Equipment, Longmen Laboratory, Luoyang 471023, China
3 Henan Key Laboratory for Machinery Design and Transmission System, Henan University of Science and Technology, Luoyang 471003, China
* Correspondence: guonan@haust.edu.cn; Tel.: +188-3979-8351

**Abstract:** Aiming to solve the problem of oil-air lubrication failure caused by the high working temperature of high-speed rolling bearings, this study proposes a method, based on the theory of gas-solid two-phase flow and bearing tribology, of predicting the dynamic temperature rise of nonlinear high-speed rolling bearings under oil-air lubrication conditions. The accuracy of the fluid–structure coupling model is verified by comparing the temperature rise test results of angular contact ball bearing at different speeds. The characteristics of oil-air lubrication circulation and the relationship between the lubrication parameters and the heat balance of the high-speed rolling bearings have been studied. The results show that the gas supply pressure of the system has a significant influence on the continuity and fluctuation of the oil film in the oil pipe nozzle. The initial rise in temperature of the inner and outer rings of the bearing and the fluid domain has a speed threshold value, and the temperature increases linearly with the bearing speed. With the increase in the oil supply and lube oil viscosity of the system, the temperature rise of the outer ring of the bearing increases first, then decreases, and finally increases again. There is an optimal oil supply 5.5 mL and optimize viscosity 68 cSt for the bearing in the work condition.

**Keywords:** rolling bearing; liquid-solid coupling; thermal equilibrium; temperature rise





## 1. Introduction

Under high-speed working conditions, rolling bearings suffer from problems such as increased friction heat generation, reduced fatigue life, cage and rolling element slippage, and rolling surface damage, which directly affect the working performance and life of the bearing system. An increase in the bearing temperature decreases the viscosity of the lubricating oil, the oil film thickness between the rolling element and the inner and outer ring raceways decreases, and the lubrication failure caused by the high temperature of the working surface leads to the failure of the spindle bearing. In addition, the temperature in the bearing system is the main factor affecting the bearing speed performance, which can be used as an important parameter to determine the bearing speed performance. Therefore, it is of great significance to study the fluid–solid coupling heat balance of high-speed rolling bearings under oil-air lubrication to investigate the bearing performance and its life.

At present, high-speed rolling bearings are widely used in railway bearings, motorized spindle bearings, engine main bearings, etc. Lubrication of a rolling bearing can efficiently reduce the heat generated by the internal friction in the bearing system. The type of lubrication method used in a rolling bearing has a significant influence on its performance under high-speed working conditions, which is the first problem to be solved in studying the heat balance of a high-speed rolling bearing.

Because of its good adhesion, sealing, lubricity and long life, grease is widely used in high-speed railway axle box bearings at home and abroad. Yang Fenglin [1] analyzed

the friction power loss of axle box bearing, calculated the thermal conductivity of bearing, established the numerical model of temperature field of axle box bearing, studied the change of temperature field of axle box bearing under different working conditions, and studied the deterioration characteristics of grease on this basis. Allmaier H et al. [2] studied the potential ways to improve the service life of grease in rolling bearings, especially as the application of rail wheel bearings, and proposed that the basic lubrication process is the key to improve the service life of rail wheel bearings. Gao P [3] established a quasi-static mechanical model of the bearing according to the working conditions, and obtained the load distribution and kinematic parameters of the bearing. The temperature distribution of the railway double row tapered roller bearing under the test conditions was studied by finite element analysis, which was consistent with the test results.

Oil and gas cooling lubrication technology has the advantages of a high DN (the product of bearing diameter (D) and speed (N)) value, good lubrication performance, low pollution, and suitability for its use in high-load, high-speed, and high-temperature bearing work. Schubring [4] studied the two-phase flow characteristics of oil-air lubrication through experiments and obtained the variation of the oil film as a function of the lubrication parameters in oil pipelines under different working conditions. Jeng et al. and Gao et al. [5–8] developed stable oil supply test equipment for oil-air lubrication and an oil-air lubrication test bench for high-speed rolling bearings. The effect of the oil pipe length, gas supply pressure, oil supply interval, and viscosity of the lubricating oil on the oil supply stability of the oil-air lubrication system was studied, and the effects of radial load, oil supply, and rotational speed on the lubrication performance in the bearings were studied. Ramesh [9,10] et al. measured the thickness of the oil film between the rolling element and the raceway of the bearing under oil-air lubrication using the capacitance method on the oil-air lubrication test bench. They studied the influence of the heat transfer mode of the bearing on its temperature field distribution under oil-air lubrication and observed that the convective heat transfer is the main heat transfer mode of the bearing under oil-air lubrication. Wu [11] studied the oil-air lubrication and preload of machine tool spindle bearings, focusing on the influence of the design parameters on the oil-air lubrication effect, and obtained the optimal oil supply parameters of oil-air lubrication. Yuan and Julong [12] studied the lubrication performance of high-speed rolling bearings at different wear stages during service under oil-air lubrication. Their research showed that when the vibration value increased to 2–2.5 times of the initial wear value, the lubrication condition of the bearing can be improved by adjusting the oil supply interval reasonably. The oil-air parameters can be adjusted reasonably according to the changes in the working conditions at different wear stages in order to obtain the best lubrication strategy for rolling bearings under various working conditions. Yan and Bei [13] studied the migration and diffusion of the oil droplets in the bearing cavity using the coupling technology of the level-set function and the volume of fluid (VOF) method, analyzed the increase in the temperature and lateral oil-air lubrication mode of the outer ring bearing at different speeds, and predicted the lubricating oil flow and heat dissipation performance in the bearing cavity and key areas.

Compared with grease lubrication, oil-air lubrication has a wide application range and is not affected by temperature. It is suitable for poor conditions such as high temperature, heavy load, high speed, very low speed, and cooling water and dirt invading the lubrication point, and the lubrication effect is obvious.

Several research studies have been done on the temperature rise characteristics of the bearing system. Cui et al. [14] established the calculation model of the oil-gas two-phase flow and revealed the relationship between the temperature rise of the bearing and the speed, oil viscosity, radial clearance, preload, and air ratio. This study showed that the air ratio has the greatest influence on the increase in the temperature of the bearing, and the air ratio has an optimal value, at which the temperature rise of the bearing is the lowest. Li et al. [15–17] established an oil-air lubrication model of rolling bearings using the computational fluid dynamics (CFD) numerical simulation method and studied the influence of the inlet flow rate on the flow uniformity under oil-air lubrication.

They observed that the uneven distribution of oil in the bearing significantly affects the temperature rise and lubrication effect of the bearing. Li et al. [18–20] used the CFD method to analyze the flow velocity, temperature distribution, volume fraction, and heat transfer coefficient distribution of a two-phase medium in a bearing chamber and studied the flow and heat transfer law of the oil and gas two-phase medium in this chamber. Their results showed that the flow velocity of the two-phase medium increases first and then decreases with the increase in the radial height, and the lubricating oil in the bearing chamber is mainly distributed in the oil return pool and the outer wall of the chamber. Zheng [21] used the optimized thermal network model to estimate the thermal performance of the angular contact ball bearings under oil-air lubrication and obtained a numerical solution using the Newton-Raphson method. The change in the bearing temperature was tested to verify the model. Their results showed that the model exhibits high accuracy. Based on the artificial neural network and genetic algorithm, Wang [22] proposed a new prediction method for the temperature rise in the bearing by using the concept of universal coupling of the oil-air lubricated angular contact ball bearings. This prediction model has exhibited good accuracy, stability, and robustness.

Harris [23] applied heat transfer technology to study the temperature field of the bearings. Based on the research by Harris, Rumbarger et al. [24] studied the temperature field of high-speed cylindrical roller bearings. However, for the convective heat transfer coefficient of bearings, Rumbarger adopted a rough physical model of 'concentric rotating rings filled with a viscous medium' to simplify the heat transfer in the bearings. This method has some errors. Palmgren [25] developed a method for calculating the friction torque of high-speed cylindrical roller bearings and the viscous torque of lubricating fluid using a large number of tests, which laid the foundation for the study of heat generation in the bearing. Takeyama et al. [26] improved Palmgren's empirical formula to a certain extent and obtained the overall power loss of high-speed cylindrical roller bearings via experimental methods. They proposed an overall method of heat generation in the bearing and its heat transfer calculation method. However, this method can only be used in cylindrical roller bearings, which have certain limitations. In the working process of the bearing, the states of the contact area of each part are very different, and the overall method cannot reflect this difference. Fang B [27] used an experimental method to study the oil-air lubrication of high-speed rolling bearings. They studied the influence of various lubrication parameters on the temperature rise of the bearings using the single parameter method and obtained the optimal lubrication parameter values for a bearing with oil-air lubrication.

In this paper, based on the theory of elastohydrodynamic lubrication (EHL), thermodynamics and tribology, the VOF (volume of fluid) model of ANSYS 19.0 software has been used to study the oil-air distribution and oil-air lubrication characteristics in an oil-air lubrication pipeline of a high-speed rolling bearing. The simulation model of the fluid-solid coupling temperature field between the high-speed rolling bearing and the fluid domain under oil-air lubrication has been established. The relationship between the oil-air lubrication parameters and the thermal balance of the temperature field has been studied, which has important research significance and practical value for improving the speed performance of the high-speed bearing.

## 2. Analysis of the Circulation Characteristics of Oil-Air Lubrication in the Bearing

### 2.1. Establishment of the Nozzle Model for the Oil-Gas Lubricated Pipeline

The oil-air two-phase flow lubrication medium forms a lubricating oil film between the rolling element of the bearing and the raceway. The formed oil film can not only separate the two friction surfaces but also can bear a certain load. The continuity and stability of oil supply in oil-air lubrication are characterized by the change in the oil film continuity and the degree of oil fluctuation in the annular flow in the oil pipeline. In this study, based on the basic theory of gas-liquid two-phase flow, the VOF model has been used to study the characteristics of annular flow in a high-speed rolling bearing oil-air lubricated pipeline.

The two-phase flow distribution, heat transfer, and temperature of the oil-air two-phase flow in the bearing chamber were calculated using a finite element numerical simulation. The geometric model of the oil pipe and nozzle, based on their connection form in the oil and gas lubrication system, is shown in Figure 1.

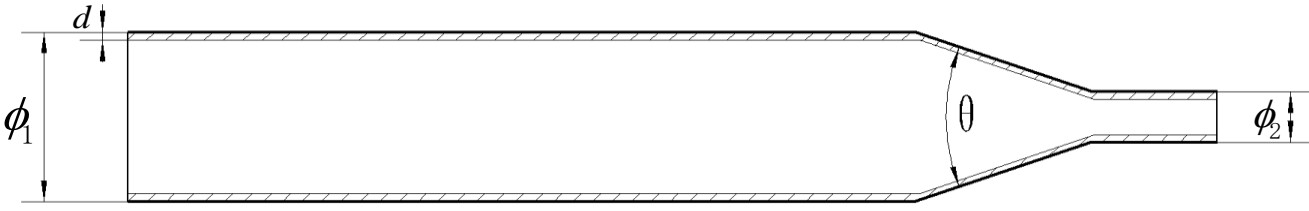

**Figure 1.** Geometric model of the oil pipeline and nozzle.

As shown in Figure 1, the pipe length was 1000 mm, the pipe diameter connected to the nozzle was 10 mm, the nozzle outlet diameter was 6 mm, and the initial oil film thickness was 0.5 mm, ignoring the effect of the nozzle length. The initial working environment in the pipeline was the standard atmospheric pressure, and considering the influence of gravity, VG68 turbine oil was selected as the lubricating oil. The density of the oil was 876 kg/m$^3$, and its dynamic viscosity was 0.058 Pa·s. The oil-air lubrication was set with air as the first phase and oil as the second phase. The gas and lubricating oil were incompressible, and there was no phase transition between the two phases.

*2.2. Analysis of the Oil-Gas Lubrication Characteristics*

In oil-air lubrication, the gas supply pressure of the oil-air system directly affects the oil-air velocity in the oil pipeline and has a huge influence on the uniformity and continuity of the oil supply at the lubrication point. The oil supply pressure of the system was set to be 3 MPa, the oil supply interval was 6 min, and the oil supplied within each interval was 5 mL. The state of the annular flow of oil inside the oil pipeline was analyzed when the gas supply pressure was 0.1, 0.3, 0.5, 0.7, and 0.9 MPa, respectively. The influence of the gas supply pressure in the system on the state of the annular flow oil film distribution in the oil pipeline and the nozzle and the change in the velocity field in the pipeline was studied.

Figure 2 shows the oil film distribution status of oil at the oil pipeline and nozzle when the gas supply pressure is 0.1 MPa. Red is the oil distribution status, and blue is the air distribution status. It can be seen from the figure that the oil can form a continuous and stable oil film on the pipe wall, and the oil at the nozzle has accumulated. On the one hand, the nozzle diameter is smaller, and on the other hand, the oil flow is relatively slow due to the smaller gas velocity under this pressure. This will not only cause oil blockage at the nozzle, but also reduce the gas volume fraction, thus reducing the speed of oil droplets when the oil leaves the nozzle. As the oil continues to block, the distribution volume of the oil leaving the nozzle is too large, which is likely to make it difficult for the oil droplets to pass through the wind curtain caused by the high-speed operation of the rolling bearing, thus reducing the lubrication efficiency.

It can be seen from Figure 3 that as the supply pressure increases, the degree of turbulence of the oil on the inner wall of the oil pipeline also increases. When the gas supply pressure is 0.3 MPa, the oil flow is stable, and a continuous oil film without any fractures is formed on the walls of the oil pipeline. When the gas supply pressure is 0.5 MPa, although the thickness of the oil film increases, the oil is continuous at the nozzle and is not atomized. Therefore, this oil will eventually enter the lubrication point in the form of separate dispersed oil droplets. When the gas supply pressure is 0.7 MPa, the oil film close to the wall fluctuates, the oil film breaks, and the oil film at the nozzle inlet appears to be mixed with oil and gas. This is mainly because the gas velocity is too large to produce atomization, and thus the lubricating oil is dispersed. When the supply pressure is 0.9 MPa, the degree of fluctuation is more prominent, the degree of turbulence of the oil increases,

the oil film fluctuates strongly, and a stable and continuous annular flow cannot be formed. Consequently, the phenomenon of oil droplet entrainment occurs in the pipeline. The oil at the nozzle position is completely atomized by air, and it is impossible to form continuous and uniform lubrication at the lubrication point. This not only reduces the utilization rate of the lubricating oil but also reduces the lubrication efficiency.

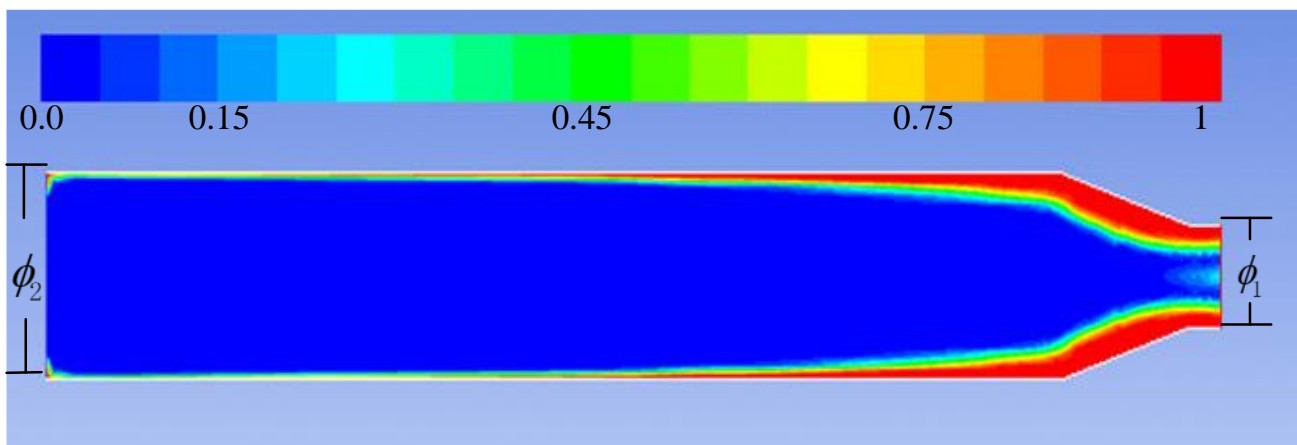

**Figure 2.** Oil-gas lubrication nozzle model and oil film distribution.

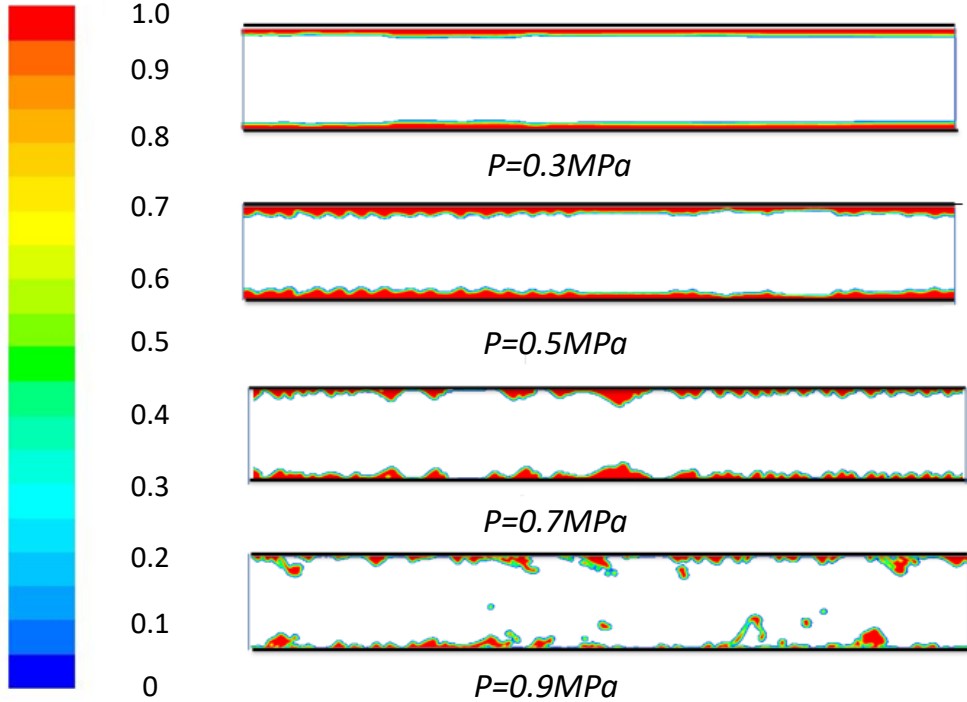

**Figure 3.** Distribution and fluctuation of the oil film in the oil pipeline and nozzle at different gas supply pressures.

### 3. Establishment of the Temperature Rise Model of the Rolling Bearing under Oil-Gas Lubrication

*3.1. Establishment of the Fluid-Solid Coupling Model for the bearing*

Taking a particular type of angular contact ball bearing as the research object, the main parameters were $\Phi130$ mm $\times$ $\Phi200$ mm $\times$ 34 mm. The fluid analysis software Ansys Fluent was used for studying the heat generation and heat transfer state of the bearing. The geometric model of the bearing is shown in Figure 4, in which Figure 4a shows the three-

dimensional model of the bearing. The bearing cavity was extracted to obtain Figure 4b. The flow field model of the bearing cavity was obtained, and the inlet of the oil and gas nozzle was added to establish the fluid–solid mixed grid model of the bearing, as shown in Figure 4c.

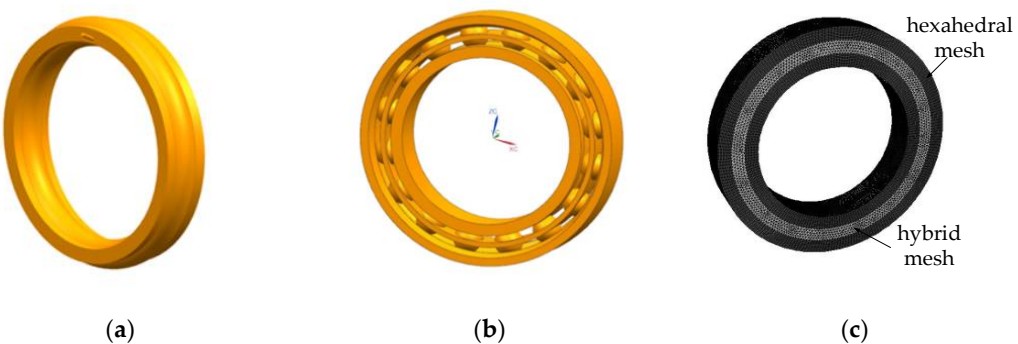

|       (a)       |       (b)       |       (c)       |

**Figure 4.** Geometric model of the angular contact ball bearing.

In the simulation, the working environment pressure was set to the standard atmospheric pressure. Considering the influence of the gravity of the bearing, the initial temperature of the bearing was 25 °C. A solver based on the pressure method, the κ-ε renormalization group turbulence model, the VOF two-phase flow model, and the heat transfer model was used. High-speed compressed gas was set as the first phase and the lubricating medium as the second phase. The oil and gas inlet was set as the pressure inlet, and the outlet pressure was set to the standard atmospheric pressure. The rotating grid model was used for the entire bearing cavity. The fluid domain and the inner ring of the bearing rotated at a particular speed. The outer ring of the bearing was relatively fixed, and its speed was zero. The iterative convergence residual of the momentum equation and the continuity equation was set to $10^{-3}$ and the iterative convergence residual of the energy equation was $10^{-7}$. In the process of high-speed rotation of the bearing, the rotation speed of the rolling element was faster. It could be approximately considered that the calorific value of each rolling element was the same. The rolling element was approximately regarded as a solid ball with a radius of $D_m/2$. The heat generation rate, q, of the bearing was calculated using the following formula:

$$q = \frac{N_f}{\pi^2 D_m (D_b/2)^2} \tag{1}$$

where q is the heat generation rate of the bearing (in w/m$^3$), $N_f$ is the internal heat of the bearing, the rolling element, $D_b$ is the diameter of the toroidal section (in mm), and $D_m$ is the diameter of the rolling element (in mm).

A heat source was applied to the rolling element and the inner and outer raceways as a set heat generation rate. Considering the convective heat transfer between the bearing and the fluid, the convective heat transfer coefficient between the bearing and the lubricating medium was calculated using the following formula:

$$\alpha_1 = 0.0986 \left[ n \left( 1 \pm \frac{D_b \cos \alpha}{D_m} \right) / v \right]^{1/2} k P_r^{1/3} \tag{2}$$

where $\alpha_1$ is the forced convection heat transfer coefficient inside the bearing, n is the bearing speed, $D_b$ is the diameter of the rolling element (in mm), $D_m$ is the average diameter of the bearing (in mm), $\alpha$ is the bearing contact angle (in rad), v is the average flow rate of the lubricant (in mm/s), k is the thermal conductivity of the lubricant, and $P_r$ is the Prandtl number.

Because the thickness of the inner and outer rings of the angular contact ball bearing is much smaller than the width of the bearing, the inner and outer rings of the bearing can be approximately regarded as the cylinder wall. Considering the heat conduction between the inner ring of the bearing and the shaft, and the outer ring and the bearing seat, the conduction thermal resistance between the inner and outer rings of the bearing, and the bearing seat and the mandrel can be expressed as

$$R_1 = \frac{\ln(d_i/d)}{2\pi\lambda_q B} \tag{3}$$

$$R_2 = \frac{\ln(D/d_0)}{2\pi\lambda_q B} \tag{4}$$

where $R_1$ is the heat conduction resistance of the inner ring, $R_2$ is the heat conduction resistance of the outer ring, d is the inner diameter of the bearing (in mm), $d_i$ is the diameter of the contact point between the inner ring raceway and the rolling element (in mm), $d_i = D_m(1 - \cos\alpha)$, $d_0$ is the diameter of the inner wall of the outer ring (in mm), D is the diameter of the outer wall of the outer ring (in mm), $d_0 = D_m + D_b\cos\alpha$, and $\lambda_q$ is the thermal conductivity of the inner and outer ring material of the bearing (in $\mathrm{w}/(\mathrm{m\cdot°C})$).

### 3.2. Experimental Verification

In order to verify the accuracy of the temperature rise simulation results of the rolling bearing, a particular type of angular contact ball bearing was used in the test bearing and the size as same as the simulation model. The lubrication method is oil and gas lubrication. In the test, the bearing was run in for one hour. When the bearing temperature dropped to room temperature, the temperature rise of the bearing at different speeds was tested.

Figure 5 shows an oil-air lubrication bearing test bench. The system was composed of three main parts: an oil-air lubrication system, a high-speed drive system, and a test piece system. The oil and gas lubrication system included an air compressor and filter, a cooling and drying device, an air storage tank, and an oil and gas generator. The high-speed drive system included a high-speed spindle, an inverter, and circulating water cooling equipment. The test piece system mainly included a test piece, an infrared thermometer, and a hydraulic loading device. In the entire test system, the air compressor continuously provided compressed air. The compressed air passed through the air filter to remove dust and particulate matter and other debris, and reached the gas storage tank. After the cooling and drying device removed the water vapor, it flowed through the oil and gas generator at a constant pressure. The oil-gas generator periodically injected lubricating oil into the oil-gas mixer in the form of oil droplets, which were transported to the lubrication point through the tubing. The frequency converter controlled the rotation speed of the high-speed spindle, realized stepless speed change, and drove the test piece through the coupling. Circulating water cooling equipment was used for cooling the motorized spindle built-in motor. A infrared thermometer was used to measure the temperature of the outer ring of the two bearings in the test piece. In the test, the main bearing was under-ring lubrication, and the test bearing was an angular contact ball bearing with oil-air lubrication.

In the experiment, the gas source was set to be 0.4–0.6 MPa, 500 L/min, the pressure of the oil and gas mixer was 1.0 MPa, and the reset pressure was 0.15 MPa. The experimental setup consisted of four distribution ports, each with a discharge of 0.015 mL/s, and the number of injections per hour was approximately 12 times (the number of injections could be changed according to the actual situation). The import tube had an inner diameter φ10 mm, a length of 2000 mm, and a horizontal spiral of 5 laps. The gas supply pressure was set to be 0.3 MPa, the oil supply interval was 6 min, the preload was 300 N, the lubricating oil viscosity was 68 cSt, and the oil supplied within each interval was 4–8 mL, respectively. When the outer ring of the test bearing was fixed and the inner ring speed changed from 3000 r/min to 10,560 r/min, the temperature rise of the outer ring of the bearing at different speeds was tested. The test results are presented below.

It can be seen from Table 1 that when the bearing speed is 3000 r/min, 4980 r/min, 8240 r/min and 10,560 r/min, respectively, compared with the test results, the simulation temperature rise error is less than 10%, indicating that the simulation model has high accuracy.

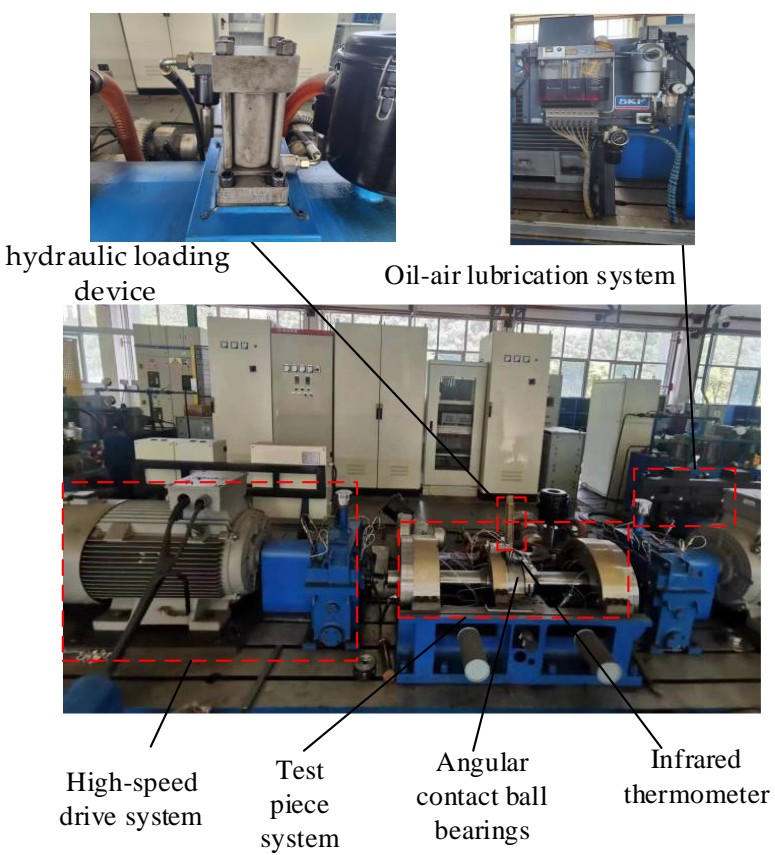

**Figure 5.** Oil-air lubrication test bench.

**Table 1.** Comparison of temperature rise test and simulation of bearing outer ring.

| Revolution Speed r/min | Test Temperature Rise °C | Simulation Temperature Rise °C | Error |
|---|---|---|---|
| 3000 | 21.7 °C | 20.5 °C | 5.5% |
| 4980 | 30.2 °C | 27.9 °C | 7.6% |
| 8240 | 40.7 °C | 37.1 °C | 8.8% |
| 10,560 | 46.6 °C | 41.9 °C | 9.7% |

## 4. Discussion of the Temperature Rise Characteristics of Rolling Bearing under Oil-Gas Lubrication

### 4.1. Influence of the Rotational Speed on the Temperature Rise Performance of the Bearing

The increase in the bearing speed aggravates the friction heat between the rolling element and the inner and outer rings, and also the change in the viscosity of the oil. Thus, the bearing speed has a significant influence on the temperature rise of the bearing. In the simulation system, the gas supply pressure was set to 0.3 MPa, the oil supply pressure was 3 MPa, the oil supply interval was 6 min, the preload was 300 N, and the viscosity of the lubricating oil was 68 cSt, as shown in Table 1. The simulation was carried out at the revolution speeds of 5000, 8000, 10,000, 12,000, 15,000, 18,000, and 20,000 *r*/min. The internal heat flow parameters of the bearing at different speeds are listed in Table 2.

**Table 2.** Internal heat flow parameters of the bearing at different rotational speeds.

| Revolution Speed r/min | Heat Generation w | Heat Production Rate w/m$^3$ | Coefficient of Convective Heat Transfer w/(m$^2 \cdot$K) | Maximum Temperature $^\circ$C |
| --- | --- | --- | --- | --- |
| 5000 | 53.4 | 125,973 | 38.62 | 35.7 |
| 8000 | 78.6 | 185,421 | 41.31 | 42.3 |
| 10,000 | 106.2 | 250,530 | 46.25 | 50.4 |
| 12,000 | 138.1 | 325,784 | 52.25 | 63.5 |
| 15,000 | 176.3 | 415,899 | 58.32 | 81.4 |
| 18,000 | 216.9 | 511,677 | 63.24 | 96.6 |
| 20,000 | 263.6 | 621,844 | 71.58 | 110.2 |

Figure 6 shows the relationship between the speed and the temperature rise of the outer ring, the inner ring, and the fluid domain of the bearing. It can be seen from the figure that the temperature rise performance of the bearing at high speed is significantly affected by the speed. The temperature rise of the inner and outer rings of the bearing and the fluid domain increases with the increase in the bearing speed. In addition, the degree of increase of these parameters also gradually increases. The speed of the bearing is proportional to the temperature rise of the inner and outer rings of the bearing and the fluid domain. The discrete points in Figure 5 were linearly fitted and empirical formulae for the temperature of the bearing at different speeds were obtained as follows:

$$Y_{inner} = -10.86878 + 0.00397x \tag{5}$$

$$Y_{outer} = -9.65789 + 0.00339x \tag{6}$$

$$Y_{fluid} = -5026667 + 0.00227x \tag{7}$$

When y = 25 $^\circ$C, the speed threshold of the outer ring, the inner ring, and the fluid domain of the bearing is 10,224, 9035, and 13,245 r/min, respectively. When the speed exceeds the threshold value, the temperature begins to rise linearly.

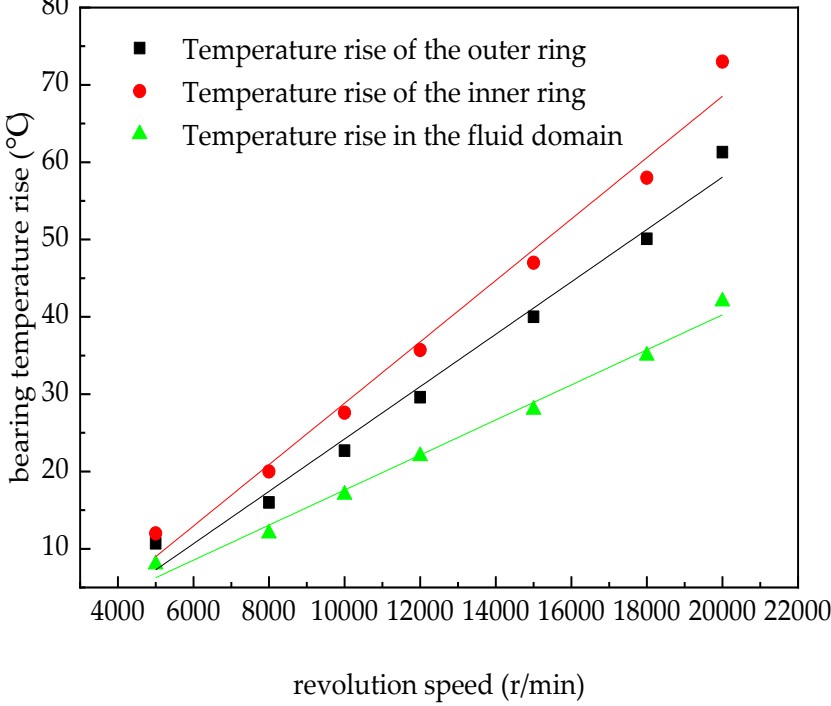

**Figure 6.** Effect of the rotational speed on the temperature rise of the bearing.

It can be seen from Figure 6 that the slope of the empirical formula for the temperature of the inner ring of the bearing is the largest and the trend of the temperature rise is the most obvious. This is because the faster the speed of the bearing, the more severe is the differential sliding between the rolling element and the inner and outer rings and the cage. The more severe the sliding friction and the rolling element spin friction, the more is the friction torque between the various parts, which leads to the increased heating of the bearing as a whole. At the same time, the internal vibration of the bearing is obvious when it rotates at high speed. Due to the oscillation of the oil film and the temperature rise of the lubricating oil, the lubrication of the bearing worsens, and some contact areas between the rolling element and the raceway are in mixed friction, resulting in a large amount of heat. The generated heat and the lubrication performance of the lubricant influence each other, resulting in increasingly poor bearing lubrication conditions and a sharp rise in the temperature of the bearing.

Under the high-speed operation of the bearing, due to the inertia force and heating, the ring will expand, resulting in a certain change in the diameter of the bearing groove bottom, which will lead to a change in the contact angle, change in the working performance of the bearing, and cause the temperature rise of the bearing. With the increase of bearing speed, the rolling component at the contact between the steel ball and the raceway increases, and the spin component of the steel ball also increases, which increases the rolling and sliding friction of the steel ball, resulting in increased heat generation and temperature of the bearing. During the operation of the bearing at high speed, the temperature of the bearing will change with the increase of the number of working cycles of the bearing. Because the bearing will generate heat when it works, the heat will also generate the temperature rise of the bearing, which will cause thermal expansion. The thermal deformation will increase, and the actual axial force of the bearing will increase due to the effect of thermal expansion. The increase of axial force further intensifies the friction heat generation. This is because, with the increase of the bearing speed, the rolling component of the contact point between the steel ball and the raceway increases, and the spin component of the contact point between the steel ball and the raceway increases. The rolling friction and sliding friction of the steel ball are increased, resulting in increased heat generation and temperature rise.

*4.2. Influence of the Preload on the Temperature Rise Performance of the Bearing*

If the bearing preload is too large, the contact stress between the bearing roller and the inner and outer tracks will increase, which will increase the friction between the two, and eventually lead to the high working temperature rise of the bearing. However, the preload is too small, and the bearing cannot run smoothly when rotating at high speed. Therefore, the preload of the bearing should be reasonably selected according to the load and service requirements of the bearing. When the operating parameters are an air supply pressure of 0.3 Mpa, oil supply pressure of 3 Mpa, oil supply interval of 6 min, rotational speed of 5000 r/min, lubricating oil viscosity of 68 cSt, and the preload is 100 N, 200 N, 300 N, 400 N, 500 N, 600 N, 700 N, respectively, the temperature field of the bearing is simulated.

It can be seen from Figure 7 that the bearing temperature rise increases linearly with the increase of preload. This is because when the bearing rotates at high speed, the friction caused by the elastic hysteresis of the bearing, the friction caused by the local differential sliding and the friction torque caused by the spin sliding of the ball along the ring groove will increase with the increase of the bearing preload, which will lead to the increase of the overall heat generation of the bearing and eventually the increase of the temperature rise of the bearing.

The bearing preload is in direct proportion to the temperature rise of the bearing outer ring. The empirical formula of the temperature rise of the bearing outer ring under different preloads can be obtained by linear fitting of discrete points:

$$Y_{temp\ rise} = 17.3 + 0.00736x$$

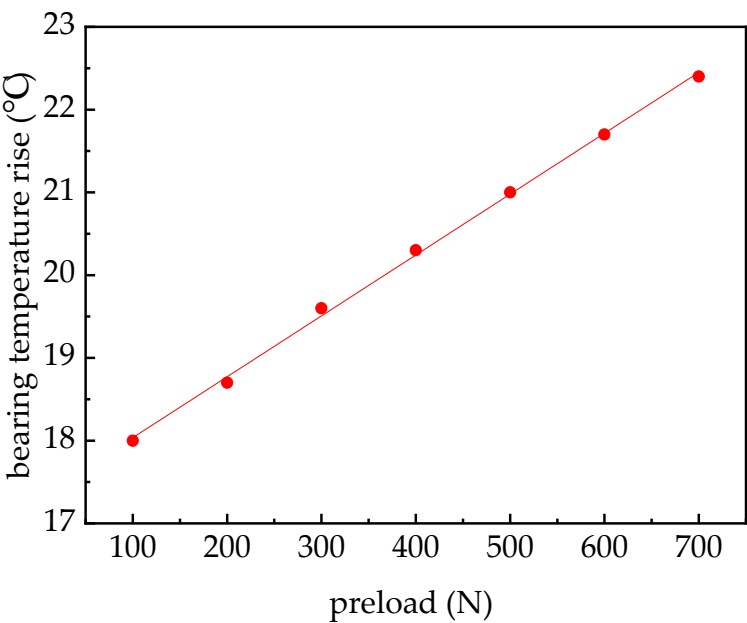

**Figure 7.** Effect of rotational preload on temperature rise of bearing.

*4.3. Analysis of the Oil Supply on the Temperature Rise Performance of the Bearing*

The lubrication oil supply plays a vital role in high-speed rolling bearings. The amount of oil supplied directly affects the thickness of the lubricating oil film within the bearing. It can not only reduce the direct contact friction heat between the roller and the inner and outer rings but can also take away a certain amount of heat in the process of circulating the oil supply. In this study, the effect of oil supply on the temperature rise of the bearing was studied. For performing this investigation, the following parameters were set: the gas pressure was 0.3 MPa, the oil supply interval was 6 min, the preload was 300 N, the viscosity of lubricating oil was 68 cSt, and the speed of the bearing was 10,000 r/min. The oil supplied within each interval of 4–8 mL was used.

Figure 8 shows the change in the temperature rise of the bearing as a function of the oil supply of the lubrication system. It can be seen from the figure that the temperature rise of the bearing increases first and then decreases with increasing oil supply, and finally increases once again. This is because at the beginning (oil supply is 4–4.5 mL), it may be in a lean state, and the friction between the roller and the inner and outer rings generates more heat. As the viscous torque of the lubricating oil increases, the heat generated by the stirring of the lubricating oil also increases; thus, increasing the temperature of the outer ring of the bearing. As the oil supply continues to increase (4.5–5.5 mL), the oil film between the bearing rolling element and the raceway gradually becomes complete, the friction heat of the bearing decreases, the heat taken away by the lubricating oil increases, and the temperature rise of the bearing decreases. In addition, the stirring friction gradually increases and reaches an equilibrium state at a certain value of oil supply. As a result, the temperature rise of the bearing is the lowest, and the oil supply is also optimum for the bearing. However, as the oil supply continues to increase (5.5–8 mL), the bearing is in an oil-rich state, and thus an excessive amount of oil is present for forming the oil film. This large amount of lubricating oil increases the heat generated due to the stirring of the bearing. When it is greater than the heat taken away after improving the lubrication conditions, the temperature rise of the bearing increases with increasing oil supply. Therefore, the change in the bearing temperature at high speed is sensitive to the change in the oil supply. When the lowest temperature rise point of the outer ring of the bearing is at approximately 5.5 mL of the oil supply, a good oil film can be formed in the bearing.

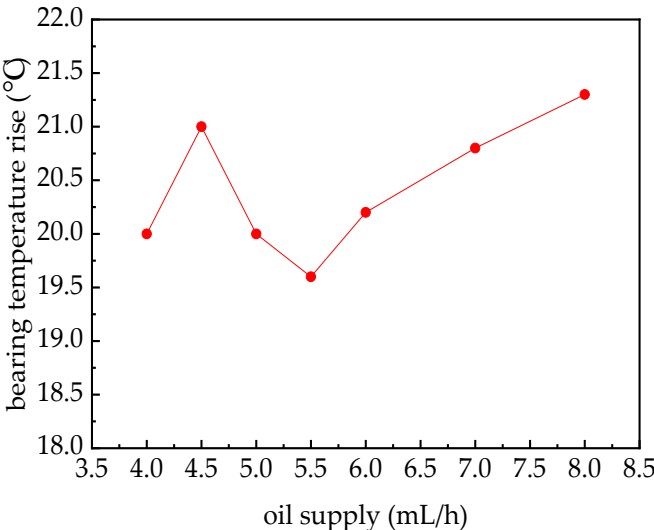

**Figure 8.** Effect of the oil supply on the temperature rise of the bearing.

*4.4. Analysis of the Lubricant Viscosity on the Temperature Rise Performance of the Bearing*

When the bearing rotates at high speed, the viscosity of the lubricating oil has a direct relationship with the oil film thickness and the heat generation of the bearing. When the bearing speed is 10,000 r/min, respectively, it maintains the air supply pressure of the lubrication system at 0.3 MPa, the preload at 300 N, and the single nozzle inlet, conduct simulation analysis on the bearing temperature rise characteristics, and obtain the influence curve of the lubricating oil viscosity on the bearing temperature rise.

It can be seen from Figure 9 that, within a certain viscosity range, the influence of lubricating oil viscosity on bearing temperature rise is small. However, when the viscosity exceeds a certain value, the bearing temperature rise gradually increases. The change trend of lubricating oil viscosity on bearing temperature rise characteristics is consistent with the theory of elastohydrodynamic lubrication. However, when the viscosity of the lubricating oil is too large, the friction torque related to the viscosity of the lubricant increases, the heat generation increases, and the temperature rise of the bearing increases. Lubricating oil with higher viscosity is easy to form oil film, but at the same time, the heat generated by viscous torque is also increasing, so the viscosity of lubricating oil should be reasonably selected according to the actual working conditions such as bearing speed and load. At the same higher speed, the lubricating oil has the best viscosity value.

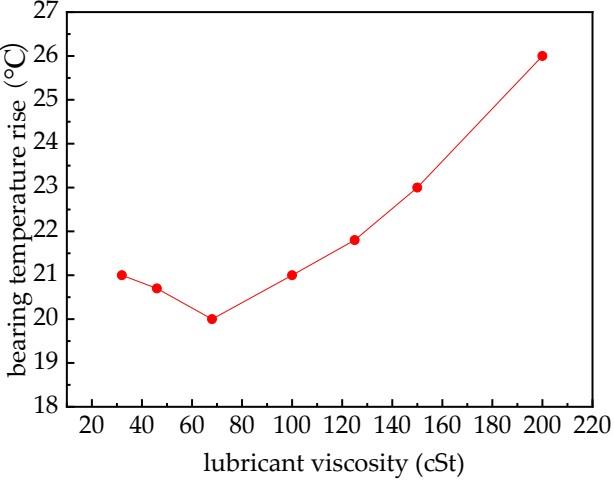

**Figure 9.** Effect of the lubricating oil viscosity on the temperature rise of the bearing.

## 5. Conclusions

In this study, based on the basic theory of the gas-liquid two-phase flow, the VOF model has been used to study the distribution of oil and gas in the oil-air lubrication pipeline of a high-speed rolling bearing. The fluid-solid coupling temperature field simulation model of a bearing under the oil-air lubrication condition has been established, and the relationship between the fluid-solid coupling heat balance of the high-speed rolling bearing under this lubrication condition has been studied. The following are the conclusions of this study:

(1) The air supply pressure in the oil-air lubrication system has a huge influence on the continuity and fluctuation of the oil film in the nozzle. If the gas pressure is too low, oil accumulation occurs in the nozzle. If the supply pressure is too large, the greater the speed of the gas in the pipeline, the more the probability of fluctuation and fracture of the oil film.

(2) The speed of rotation has a great influence on the temperature rise of high-speed rolling bearing. With the increase of the speed shaft, the temperature rise of the bearing inner ring is faster than that of the bearing outer ring and the bearing inner fluid domain. The temperature rise of the three increases linearly with the bearing speed, and the initial temperature rise has a threshold value of rotation speed. In addition, with the increase of preload, the temperature rise of bearing outer ring also shows a linear increase trend.

(3) In high-speed rolling bearings, with the increase of oil supply and lubricating oil viscosity, the bearing temperature rise decreases first and then increases. The system oil supply and lubricating oil viscosity have a certain impact on the temperature rise of high-speed rolling bearings, and there is an optimal oil supply and viscosity value.

**Author Contributions:** Author Contributions: Conceptualization, M.Z.; methodology, M.Z. and N.G.; software, M.Z.; validation, X.M. and M.Z.; investigation, F.Y.; resources, F.Y.; data curation, N.G.; writing—original draft preparation, M.Z.; writing—review and editing, R.G.; supervision, Y.X.; project administration, X.M.; funding acquisition, X.M. All authors have read and agreed to the published version of the manuscript.

**Funding:** This research was funded by National Key R & D Program of China (Grant No.2021YFB2011000); Henan Science and Technology Project (212102210365); Research Program in University of Henan Province (Grant No. 21A460014) and Major Science and Technology Project of Henan Province (221100220100).

**Data Availability Statement:** Data is contained within the article.

**Conflicts of Interest:** The authors declare no conflict of interest.

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
