# Peer review of "Analysis of Circulation Characteristics and Heat Balance of High-Speed Rolling Bearing under Oil-Air Lubrication"

_lubricants, doi:10.3390/lubricants11030136_

Round 1
Reviewer 1 Report
The paper in its current form lacks many important details to make it more clear to readers. Some suggestions are as follows:
(1) The specific application for this study is not mentioned or explained. Is this work done for specific type of bearings? What about bearings with a constant volume of lubricant?
(2) Are the results transferable to other bearing types or applications? For example, are the results transferable to railroad bearings that have a predetermined volume of lubricant?
(3) The behavior described in the paper is similar to what happens in railroad bearings yet there is no mention of any such results. There are many papers that have been written detailing bearing operating temperatures under several operating speeds yet none of those are referenced or compared to the results presented in this paper.
(4) The experimental setup is poorly described. A schematic diagram showing the location of the temperature measurements and how they were taken can greatly improve the study. As it is, Figure 9 is not very descriptive or clear.
(5) Are the simulated results obtained from an equation or the CFD? It is not clear in the paper.
(6) Has the effect of the ambient temperature been studied or is it irrelevant for this application? When the lubricant is cold, its viscosity is higher leading to frictional heating when the bearing is suddenly started from a cold start.
(7) How do the results obtained improve the specific application. Meaning, how can these results be used to improve the operation of these bearings? Without an application in mind, are these results just general results to be considered?
(8) A better motivation for the work done would greatly improve the paper. In other words, the need for this work is not described well or highlighted in the paper.
(9) What is the maximum operating temperature of the lubricant being modeled. At 110 degrees C, the lubricant is not that effective depending on the lubricant type. A statement to that aspect can go a long way towards describing the application type.
(10) The linear increase in temperature shown in Fig 5 can be supported with other results published showing similar trends. Try looking up some railroad bearing temperature studies performed. That can add value to your literature review.
Reviewer 2 Report
Comments to the authors:
1. First line of the abstract needs to be revised and it is not clear.
2. The optimal oil supply has to be mentioned in the abstract.
3. The research gap and novelty of the work are not clear in the manuscript.
4. Quality of Fig 2 and 3 needs to be enhanced.
5. ANSYS software version needs to be mentioned.
6. All the equations have to be cited properly.
7. Figure 5 has to be discussed more with literature support.
8. In Figure 6, the temperature was decreased at 5.5 ml/h fuel rate. Why?
9. A separate discussion section is mandatory and all the obtained results have to be discussed clearly.
10 Ref 2 is not in the proper format.
11 Few older references can be replaced with the latest ones.
Round 2
Reviewer 1 Report
Paper is much improved with additions
Reviewer 2 Report
All the best to the authors